# Path Planning in Threat Environment for UUV with Non-Uniform Radiation Pattern

**DOI:** 10.3390/s20072076

**Published:** 2020-04-07

**Authors:** Andrey A. Galyaev, Alexander V. Dobrovidov, Pavel V. Lysenko, Mikhail E. Shaikin, Victor P. Yakhno

**Affiliations:** Institute of Control Sciences of RAS, Moscow 117997, Russia; dobrovidov@gmail.com (A.V.D.); pashlys@yandex.ru (P.V.L.); shaikin@ipu.ru (M.E.S.); vic-iakhno@mail.ru (V.P.Y.)

**Keywords:** UUV path planning, non-detection probability, passive and active surveillance in the acoustic field, non-uniform radiation pattern

## Abstract

The problem of optimal trajectory planning of the unmanned underwater vehicle (UUV) is considered and analytically solved. The task is to minimize the risk of detection of the moving object by a static sonar while moving between two given points on a plane. The detection is based on the primary acoustic field radiated by the object with a non-uniform radiation pattern. In the first part of the article, the probability of non-detection is derived. Further, it is used as an optimization criterion. The non-uniform radiation pattern of the object differentiates this work from previous research in the area. The optimal trajectory and velocity law of the moving object are found, as well as the criterion value on it.

## 1. Introduction

The extensive development of unmanned mobile vehicles has led to a new area of control problems known as “path planning in the threat environment.” The threat environment is defined as a set of agents (they are called conflicting) that the controlled vehicle must not approach while performing its main task. The control goal is to minimize the negative impact of conflicting agents by selecting the route, parameters of its movement and/or operating modes of technical devices. Depending on the nature of the specific task, the objectives of the negative impact can be considered to be the detection of an object, the approach to the conflicting object to the distances from which it is possible to defeat, and so forth. In the problem of evasion the moving object has to remain covert for the detector system [1]. These problems are extremely relevant for modern-day warfare applications. The technical progress allows using of non-trivial onboard algorithms for unmanned vehicles from wide scientific areas such as optimal control and differential games.

In Reference [2], the problem of automated path planning of combat unmanned aerial vehicles (UAVs) in the presence of radar-controlled surface-to-air missiles is considered. This task includes the interaction of three subsystems—the aircraft and its characteristics, the radar and its capabilities, and the missile and its striking properties. The paper proposes an approach to a model consisting of three subsystems united in a single system. In general, the literature on UAV path planning can be divided into two groups. The first group considers the problem under the assumption of isotropic radar capabilities (i.e., the independence of the reflected signal from the UAV’s spatial orientation), and the second group assumes the implementation of such dependence in the model.

Mathematical criteria for the success of missions performed by moving objects can be the following—the probability of rescue, the motion time on the trajectory, the mathematical expectation of the time interval until the first detection on the path, the path length, and so forth [3,4,5,6,7,8,9]. These criteria are often in conflict with each other, that is, they are contradictory. Some of them may be functionally related. In References [10,11], the path planning problem criterion is based on the artificial potential field which is formed by the obstacles. In Reference [12], optimal path consists of lines and circular arcs (2D Dubins curves). A fairly complete overview of path planning algorithms for the unmanned underwater vehicle (UUV) finding an optimal or sub-optimal route between an origin point and a termination point in marine environments is given in Reference [13].

The solution of the problem of hidden penetration of UUV into the protected area is carried out in the conditions of network-centric counteraction of the opposite side, equipped by means of illumination of the underwater situation and the detection of UUV. Unlike previous works [1,4,5,8], the acoustic signal emitted by a moving object has a non-uniform pattern. One example of a hydroacoustic signal detector is a stationary sonar system (SSS). In the case of a passive mode of reception, the SSS detects UUV using a radiated signal generated by UUV motion [14]. However, in the case of an active mode, UUV reveals itself by the secondary field of the reflected probing signal. An indicator of the effectiveness of a hidden object penetration into the area is the probability of UUV non-detection by means of SSS [3,15]. In the problem of penetration, the maximal UUV invisibility is achieved by finding the route and the law of speed change on it, at which the maximum of this probability is reached. On the other hand, an indicator of the effectiveness of the detection by SSS is the probability of detection of a moving UUV in the conditions of receiving a signal against the background of random noise of the aquatic environment. Therefore, the problem of a hidden penetration is naturally formulated as the problem of testing two statistical hypotheses: $H_{0}$ (there is no UUV object in a given area) and $H_{1}$ (there is an object in the area). Both hypotheses are complex. They depend on the trajectory of the object and its speed.

The choice of the route with the maximum covertness is the subject of a certain optimization problem. Route selection is a difficult task for the calculus of variations and since it is also posed as a probabilistic task, a probabilistic measure on the functional space of the trajectories must be specified, for example, on the space of continuous functions C(0,T). Articles [16,17,18,19] are devoted to the problems of the optimal planning of the UUV route under threat conditions, in which other probabilistic and energy criteria are implemented.

The noted features of the UUV route optimization problem determine the structure of the proposed work. The probabilistic properties of the environment in which the UUV passes do not depend on the UUV, they are set a priori and are determined by the effectiveness of the opposing side, which is SSS. Therefore, Section 2 of the work contains information about the proposed principle of operation of the detection station. The main theme here is the feature of a digital processing of broadband sonar signals received by its linear antenna arrays.

Section 2 discusses the problem of calculating the probabilities of UUV detection in both passive and active observation modes. One of the options for setting the probabilistic distribution of the acoustic noise field, which determines the degree of visibility of the UUV for the SSS, is as follows. The entire tracking area is divided into many spatial zones, in each of which the SSS can calculate (and UUV—estimate by modeling) the probability of detection of an extraneous object. The threat map obtained in this way then serves as initial information for calculating the total probability of detection on the set of zones through which any selected or given object trajectory runs. This probability is identified with the probability of detection on the trajectory. Section 2 also considers the case of detection with a small signal to noise ratio. At the same time, it is useful to use important concepts of the risk of detection and the functionality of threats. Section 3 completes the work, where a more general optimization problem is posed, in which, in addition to the antenna array radiation pattern, it is possible to take into account the scattering indicatrix of the noise-emitting object as a function of its course angle. The choice of the course angle in the task of optimizing the trajectory allows one to improve the indicators of UUV covertness. In Section 4 the obtained solutions of the optimal path planning problem are discussed and compared with known results. Section 5 gives the direction for future work.

## 2. UUV Non-Detection Probability

### 2.1. Features of the Reception of Broadband Signals by Antenna Arrays

An antenna array with $N_{H}$ receiving elements is a multi-channel receiving device with output signals $x_{i}\left(t\right),\;i=1,\ldots ,N_{H}$, whose vector x(t) is represented in the time and frequency domains by the relations
(1)$$\begin{array}{c} x\left(t\right)=\int_{-\infty }^{t}h\left(t-\tau \right)s\left(\tau \right)d\tau +\xi \left(t\right),\;\\ X(\omega )=M(\omega )S(\omega )+\xi (\omega ). \end{array}$$

The matrix function h(·) characterizes the effects of the propagation of the signal s(t) generated by the source, as well as the effects of possible distortions introduced by the receiving elements. The front of the wave received in the far zone is considered flat, and then the signal s(t) becomes a scalar function of time. The wave front does not reach the receiving elements simultaneously, but with delays $\tau_{i}=\left(n\cdot r_{i}\right)/c$, where n is the unit direction vector of the wave front, $r_{i}$ is the coordinate vector of the *i*-th antenna element and *c* is the signal propagation speed. In a non-dispersive medium the matrix h(·) becomes a vector with components $h_{i}\left(t\right)=\delta \left(t-\tau_{i}\right)$, that reflect only the time delay.

In the two-dimensional problem, the vectors n,r are located in the same plane and it is assumed that the reception is carried out on a linear equidistant lattice with an inter-element distance *d* located along the axis Ox, and the angle between n and the axis Oy is equal θ. The signals from various array elements are weighed, generally speaking, with complex weights $w_{i}$, $i=1,\ldots ,N_{H}$ and summed. The frequency response of such an antenna array has the form
$$\begin{array}{c} H\left(j\omega ,\theta \right)=\sum_{i=1}^{N_{H}}w_{i}e^{-j(i-1)\psi },\;\\ \psi =2\pi (d/\lambda )\sin\theta ,\;\lambda =2\pi c/\omega . \end{array}$$

If the array is tuned to the frequency $\omega_{0}$ of the received signal then the function $H\left(j\omega_{0},\theta \right)=A\left(\theta \right)=\sum_{i=1}^{N_{H}}w_{i}e^{-j(i-1)\psi }$ depends on the angle θ. Moreover, it determines the normalized radiation pattern of the antenna array G(θ)=$10\log(|A\left(\theta \right){|}^{2}/N_{H}^{{}^{2}})$ (in terms of power). In a dispersive medium, the signal s(t) in Equation (Equation 1) is a vector with a component $s_{i}\left(t\right)$ at the input of the *i*-th element of antenna.

If the signal $s_{i}\left(t\right)$ has a carrier frequency $f_{i}\neq f_{0},\;\;i=1,\ldots ,N_{H}$, then it is impossible, generally speaking, to tune the array simultaneously to each of these signal frequencies. This fact determines the difference in the methods of receiving narrow-band and wide-band signals on the antenna arrays. During broadband reception, the signal $s_{i}\left(t\right)$ received by the *i*-th antenna element is fed to a transverse filter with amplitude and phase frequency characteristics that are adjustable for it, wherein each channel has its own transverse filter. If the latter is synthesized on the basis of a multi-tap delay line (with a delay $\Delta_{i}$ between adjacent taps), the choice of value $\Delta_{i}$ depends on the frequency sub-band of the signal $s_{i}\left(t\right)$ that this filter processes.

Consider some channel of the array that processes a frequency-limited portion [−F,F] of the input signal spectrum. The Fourier transform S(f) of the output signal of such a filter has a limited support. By Kotelnikov’s sampling theorem, a discrete set of points $s\left(i\right):=s\left(t_{i}\right)$, defined by the formula $s\left(i\right)=\int_{-F}^{F}S\left(f\right)e^{j2\pi f\;i\Delta t}df\;$(Δtis the time interval between adjacent sample points), is enough to restore the function from these points. Namely, just put $S\left(f\right)=\Delta t\sum_{i=-\infty }^{\infty }s\left(i\right)e^{-j2\pi fi\Delta t}$. Similarly, the function s(t) can be reconstructed from a discrete sequence s(i) according to the same theorem. In particular, if t=kΔt, we have s(kΔt)=Δts(k)2F=s(k). At intermediate points, the use of an expression of the type sint/t serves as an interpolation scheme which restores s(t) by values s(i).

For digital computers, the use of the Kotelnikov’s theorem is not entirely convenient. It would be desirable to have a slightly different form of the Fourier transform, in which they are also specified discretely, but the limits of summation are finite both in the direct and inverse Fourier transforms. This form is used in the theory of the fast Fourier transform (FFT), where the time sequence $s\left(i\right),\;\;i=0,1,\ldots ,N_{0}-1$ and its frequency FFT sequence $S\left(k\right),\;\;k=0,1,\ldots ,N_{0}-1$ are set by formulas
$$S\left(k\right)=\Delta t\sum_{i=0}^{N_{0}-1}s\left(i\right)e^{-j\frac{2\pi ik}{N_{0}}},\;s\left(i\right)=\frac{1}{N_{0}\Delta t}\sum_{k=0}^{N_{0}-1}S\left(k\right)\;e^{j\frac{2\pi ik}{N_{0}}},\;$$
where $S\left(k\right):=S\left(f_{k}\right),\;\;\;\;f_{k}=\frac{k}{N_{0}\Delta t}$.

The FFT algorithm can be given in a matrix notation [20]
(2)S=Ws,
where s is a vector representation of the sequence $s\left(i\right),\;\;i=0,1,\ldots ,N_{0}-1$, S is a vector representation of the sequence $S\left(k\right),\;\;k=0,1,\ldots ,N_{0}-1$ and W is a unitary matrix with elements
$$W_{N_{0}}^{ik}=\exp\left\{-j\frac{2\pi }{N_{0}}ik\right\}.$$

From this fact, the equivalence of the results obtained using the time and frequency representations of the data follows. In particular, if s is a random vector of Gaussian time samples, then the vector S in Equation (Equation 2) of frequency samples of its Fourier image will also be Gaussian. Recalculation of covariance matrices from one representation to another is simplified due to the unitarity of the transition matrix. In this paper, both representations are used equally. In particular, such an important parameter in the problems of detecting and evaluating signals as the signal-to-noise ratio is easily recalculated.

Finally, the last remark concerns the inclusion of scattering indicatrix (in power) for an object emitting a signal. The indicatrix depends on the course angle of the object. Multiplying the scattering indicatrix and the radiation pattern of the receiving antenna, we obtain a function W(ω,θ,ϕ) depending on the frequency and two angular variables, which determines the strength of the target at the receiving point. It can be shown that for the case of a spatial (and not just flat) problem, the function W(ω,θ,ϕ) can be represented by the Chebyshev polynomials with coefficients $g_{mn}\left(\omega \right)$ depending on the frequency. The coefficients $g_{mn}\left(\omega \right)$ are downloaded in the form of tables from the database for the entire set of operating frequencies of the monitoring station.

The selection of weighting coefficients of antenna arrays and frequency channel parameters for a broadband signal processing is not considered in the work. As noted in the introduction, the purpose of the work is not to optimize the detection procedure, but to optimize the trajectory of the object in order to evade detection.

### 2.2. Non-Detection Probability of UUV under Passive Surveillance

The covertness of the UUV on the selected trajectory with a given velocity law can be characterized by the probability that during the passage of the whole route it will not be detected at any tact. Such probability is denoted by $P_{nd}$ and will be called the non-detection probability of UUV on the trajectory. Let us denote as $T_{0}$ the UUV transit time of the whole trajectory, and the duration of the tact will be ΔT. Then the entire trajectory can be divided into $N=\frac{T_{0}}{\Delta T}$ segments, on each of which a decision is made about the absence of an object and the corresponding probabilities $P_{nd}^{\left(j\right)},j=1,\ldots ,N$ are calculated. Then, assuming statistical independence of the signals on each segment, one of the simplest variants of the probability measure on the trajectory is defined as the product of the probabilistic measures on the segments of the trajectory
(3)$$P_{nd}=\prod_{j=1}^{N}P_{nd}^{j}.$$

In the case of several SSS, the product of expressions of the form Equation (Equation 3) over all available SSS is taken [2]. Thus, the criterion in the formalization problem of the stealth concept of UUV is the Formula (Equation 3). Now the task is to construct the optimal trajectory and the optimal UUV velocity law, maximizing the UUV non-detection probability $P_{nd}$.

In passive mode, the signal on the elements of the antenna array is described by the formula
$$x\left(t\right)=\delta \cdot s\left(t\right)+\xi \left(t\right),\;\;t=1,\ldots ,N_{T},$$
where x(t), s(t), ξ(t) are Gaussian input signal, object signal and noise, respectively, and $N_{T}=\frac{\Delta T}{\Delta t}$ – number of discrete time samples. Here δ=0 in the case of hypothesis $H_{0}$ (a signal from UUV is absent) and δ=1 in the case of alternative $H_{1}$ (a signal from UUV is present).

If the bearing (with the angle θ) to the UUV does not coincide with the normal to the antenna line and it is known, then the hydrophone input signal is
$$x^{\left(k\right)}\left(t\right)=s\left(t-\left(k-1\right)d\;\sin\theta /c\right)+\xi^{\left(k\right)},\;\;\;k=1,\ldots ,N_{H},$$
where *d* is the distance between antenna elements, *c* is a speed of sound in water and $N_{H}$ is the number of hydrophones. This signal at the *k*-th hydrophone is a function of time and after sampling in time it turns into a vector
$$y^{\left(k\right)}=\left(y^{\left(k\right)}\left(t_{1}\right),y^{\left(k\right)}\left(t_{2}\right),\ldots ,y^{\left(k\right)}\left(t_{i}\right),\ldots ,y^{\left(k\right)}\left(t_{N_{T}}\right)\right)$$
of size $N_{T}$. The covariance matrix of this vector is equal to the sum of the covariance matrices $K_{s}+K$ of the signal and noise respectively, where, due to the independence of time samples of noise, the matrix becomes diagonal, and the matrix of the signal from the object, due to the assumed stationarity, is a Toeplitz matrix with elements
$$k_{s,(l,k)}^{t}\left(i\right)=E\left[s\left(t_{i}+l\;\Delta \tau \right)s\left(t_{i}+k\;\Delta \tau +i\Delta t\right)\right],$$
where a pair of indices (l,k) fixes the numbers of hydrophones, and iΔt corresponds to the time shift of the signals during sampling by time in step $\Delta t=t_{i+1}-t_{i}$. Similarly, Δτ=dsinθ/c means the inter-element delay in the equidistant linear array when the wave is incident at an angle θ. The time sequence of $N_{T}$ vectors in the number of $N_{H}$ pieces can be represented as a single column vector
$$Y_{t}={\left(y^{\left(1\right)},y^{\left(2\right)},\ldots ,y^{\left(N_{H}\right)}\right)}^{T}$$
with $N_{T}N_{H}$ elements. In modern SSS with linear antenna array the incoming signal is often preprocessed in the receiving path using FFT. Consider the detector, which performs FFT of time-sampled signals coming from the hydrophones of the linear antenna. As a result, we get a sample of Gaussian centered complex random vectors $x\left(f\right),\;f=1,\ldots ,N_{F}$ of dimension equal to the number of hydrophones $N_{H}$ in antenna. Further processing of the received signal is assumed not in the time but in the frequency domain, since in the case of broadband reception, it is possible to select the frequency ranges most informative for the received useful signal. Therefore, the FFT is applied to the temporal components of the vector $Y_{t}$.

After the FFT conversion, the observation signal is described as
$$Y_{f}={\left(Y^{\left(1\right)}\left(f\right),\ldots ,Y^{\left(N_{H}\right)}\left(f\right)\right)}^{T},\;\;\;f=1,\ldots ,N_{F}.$$

The detector works according to the threshold principle, according to which the observations obtained at one tact are converted into decisive statistics compared with the threshold. If the threshold is exceeded, a decision is made about the presence of the object. A false alarm is when the threshold is exceeded in the absence of an object. Such an event is random, and the threshold is selected from the condition that the probability of false alarm is equal to a given small number α(0<α<1).

After pre-processing at each hydrophone (band-pass filtering, time-shift of the input signal, sampling by time of the continuous signal and FFT of them), the input signal to be decided is a set of random vector-columns $Y^{\left(k\right)}\left(f\right),\;\;k=1,\ldots ,N_{F}$ with a dimension equal to the number of hydrophones $N_{H}$. Moreover, $Y^{\left(k\right)}=\xi^{\left(k\right)}$ in the case of a hypothesis and $Y^{\left(k\right)}=S^{\left(k\right)}+\xi^{\left(k\right)}$ in the case of an alternative. A $\left(N_{F}N_{H}\right)$-vector $Y=Y_{f}$ is compiled from complex vectors $Y^{\left(1\right)},\ldots ,Y^{\left(N_{H}\right)}$. Then the likelihood ratio for Gaussian vector Y can be determined as [21]
(4)$$\Lambda =\sqrt{\frac{\det K_{\xi }}{\det(K_{s}+K_{\xi })}\exp\left\{\frac{1}{2}Y^{*}\left[K_{\xi }{}^{-1}-{\left(K_{\sigma }+K_{\xi }\right)}^{-1}\right]Y\right\}},$$
where $K_{\xi }$ is the covariance matrix of a random vector Y of the frequency components of the signal in the case of a hypothesis $H_{0}$, and $K_{\sigma }+K_{\xi }$ is the covariance matrix of the vector in the case of an alternative $H_{1}$. Assuming noise independence at various hydrophones, covariance matrices can be represented in a block form
$$K_{\xi }=\left(\begin{array}{c} K\;O\;\;O\;\;.\;\;.\;\;.\;\;O\\ O\;\;K\;O\;\;.\;\;.\;\;.\;\;O\\ .\;\;.\;\;.\;\;.\;\;.\;\;.\;\;.\;\;.\;\;.\;\;.\;\;.\;\\ O\;\;O\;\;\;O\;\;.\;\;.\;\;.\;\;K \end{array}\right),\;\;K_{\sigma }=\left(\begin{array}{c} K_{s}^{11}\;\;\;\;K_{s}^{12}\;\;.\;\;.\;\;.\;\;.\;\;.\;K_{s}^{{1N}_{F}}\\ K_{s}^{21}\;\;\;\;K_{s}^{22}\;\;.\;\;.\;\;.\;\;.\;\;.\;K_{s}^{{2N}_{F}}\\ .\;\;.\;\;.\;\;.\;\;.\;\;.\;\;.\;\;.\;\;.\;\;.\;\;.\;\;\;.\;\;\;.\\ K_{s}^{N_{F}1}\;K_{s}^{N_{F}2}\;\;.\;\;.\;\;.\;K_{s}^{N_{F}N_{F}} \end{array}\right),$$
where $K_{}$ is the covariance matrix of a noise which is the same on each of the frequency channels of the antenna array processor, and $K_{s}^{IJ}$ is the Toeplitz covariance matrix of the vector signal s(f) from an object with elements $k_{s,(l,k)}^{f}\left(i\right)=E\left[s\left(f_{i}+l\;\Delta u\right)s\left(f_{i}+k\;\Delta u+i\Delta f\right)\right],$ where Δf=1/Δt,Δu=1/Δτ and *O* is the square zero matrix. For simplicity we assume that signal s is stationary. Then the covariance matrix of signal s can be represented as $K_{\sigma }=\sigma_{s}^{2}R_{s}$, where $R_{s}$ is the normalized covariance matrix, and $\sigma_{s}^{2}$ is the variance of the signal from the object.

### 2.3. Non-Detection Probability of UUV under Passive Surveillance at a Small Signal/Noise Ratio

The Hermitian form in the likelihood ratio Equation (Equation 4) is reduced to
$$Y^{*}[K_{\xi }^{-1}-{\left(K_{\sigma }+K_{\xi }\right)}^{-1}]Y=Y^{*}(K_{\xi }^{-1}K_{\sigma }K_{\xi }^{-1}{\left(I+K_{\sigma }K_{\xi }^{-1}\right)}^{-1}Y,$$
which allows in the case of a small signal/noise ratio to significantly simplify this expression, bringing it to the form
$$Q\left(Y\right)=Y^{*}\left(K_{\xi }^{-1}K_{\sigma }K_{\xi }^{-1}\right)Y.$$

Then in the far detection zone the likelihood ratio is approximated as a function of the statistics [22]
(5)$$Q=\sum_{k=1}^{N_{H}}\sum_{f=1}^{N_{F}}{Y^{\left(k\right)}}^{*}\left(f\right)W\left(f\right)Y^{\left(k\right)},$$
where $W\left(f\right)=K_{\xi }^{-1}\left(f\right)R_{s}\left(f\right)K_{\xi }^{-1}\left(f\right)$.

In this case, the likelihood ratio is a monotonically non-decreasing function of statistics *Q* and there is a uniformly the most powerful criterion for testing the hypothesis $H_{0}$ against $H_{1}$. This criterion is to compare the statistics *Q* from Equation (Equation 5) with the threshold *h* chosen from the condition that the probability of false alarm equals to a given number α: $P(Q>h|H_{0})=\alpha$.

By virtue of the central limit theorem the probability distribution of the statistics in Equation (Equation 5) is approximately normal with expectation $E[Q|H_{0}]$ and variance $D[Q|H_{0}]$ in the case of the hypothesis $H_{0}$ (there is no useful signal) and with expectation $E[Q|H_{1}]$ and variance $D[Q|H_{1}]$ in the case of the alternative $H_{1}$. When calculating the UUV probability of detection it is assumed that the bearing on the UUV is known. In this case, the probability of UUV being undetected in the next cycle is obtained as
(6)$$\begin{array}{c} P\left(Q\leq h|H_{1}\right)=\Phi \left(\frac{\sqrt{D[Q|H_{0}]}}{\sqrt{D[Q|H_{1}]}}\Phi^{-1}\left(1-\alpha \right)-\frac{E\left[Q|H_{1}\right]-E\left[Q|H_{0}\right]}{\sqrt{D[Q|H_{1}]}}\right), \end{array}$$
where Φ is a function of the standard normal distribution $\Phi \left(x\right)=\frac{1}{\sqrt{2\pi }}\int_{-\infty }^{x}\exp\left(-t^{2}/2\right)dt$.

Due to Equation (Equation 5) the next expressions for distribution of statistics *Q* moments are valid
(7)$$E\left[Q|H_{1}\right]-E\left[Q|H_{0}\right]=\sum_{f=1}^{N_{F}}\mathrm{tr}[W\left(f\right)\sigma_{s}^{2}R_{s}\left(f\right)],$$
(8)$$D\left[Q|H_{0}\right]=2\sum_{f=1}^{N_{F}}\mathrm{tr}[W\left(f\right)K_{\xi }\left(f\right){]}^{2},$$
(9)$$D\left[Q|H_{1}\right]=2\sum_{f=1}^{N_{F}}\mathrm{tr}[W\left(f\right)\left(\sigma_{s}^{2}R_{s}\left(f\right)+K_{\xi }\left(f\right)\right){]}^{2}.$$

Taking into account Equations (Equation 7) – (Equation 9) the Formula (Equation 6) is representable in the form
$$\begin{array}{c} P\left(Q\leq h|H_{1}\right)=\Phi \left(\begin{array}{c} \frac{\Phi^{-1}\left(1-\alpha \right)\sqrt{\sum_{f=1}^{N_{F}}\mathrm{tr}[W\left(f\right)K_{\xi }{\left(f\right)]}^{2}}}{\sqrt{\sum_{f=1}^{N_{F}}\mathrm{tr}{\left[W\left(f\right)(\sigma_{s}^{2}R_{s}\left(f\right)+K_{\xi }\left(f\right))\right]}^{2}}}-\frac{\sum_{f=1}^{N_{F}}\mathrm{tr}[W\left(f\right)\sigma_{s}^{2}R_{s}\left(f\right)]}{\sqrt{2\sum_{f=1}^{N_{F}}\mathrm{tr}[W\left(f\right)\left(\sigma_{s}^{2}R_{s}\left(f\right)+K_{\xi }\left(f\right)\right){]}^{2}}} \end{array}\right). \end{array}$$

In the absence of information about covariances between hydrophones we accept $K_{\xi }\left(f\right)=\sigma_{\xi }^{2}I,\;R_{s}\left(f\right)=I$ (*I* is identity matrix), and then
(10)$$P\left(Q\leq h|H_{1}\right)=\Phi \left(\frac{\sigma_{\xi }^{2}\Phi^{-1}\left(1-\alpha \right)}{\sigma_{s}^{2}+\sigma_{\xi }^{2}}-\frac{\sigma_{s}^{2}\sqrt{N_{F}N_{H}}}{\left(\sigma_{s}^{2}+\sigma_{\xi }^{2}\right)\sqrt{2}}\right).$$

Finally, the UUV non-detection probability on the trajectory is equal (with a small ratio signal/noise)
(11)$$\begin{array}{c} P_{\mathrm{nd}}=\prod_{j}P\left(Q_{j}\leq h|H_{1}\right)=\prod_{j}\Phi \left(\frac{\sigma_{\xi }^{2}\left(j\right)\Phi^{-1}\left(1-\alpha \right)}{\sigma_{s}^{2}\left(j\right)+\sigma_{\xi }^{2}\left(j\right)}-\frac{\sigma_{s}^{2}\left(j\right)\sqrt{N_{F}N_{H}}}{\sqrt{2}(\sigma_{s}^{2}\left(j\right)+\sigma_{\xi }^{2}\left(j\right))}\right). \end{array}$$

The next lemma is valid.

**Lemma** **1.**
*On the i-th cycle of observation non-detection probability of UUV under passive surveillance at a small signal/noise ratio can be approximately formulated as*
$$\begin{array}{c} P_{\mathrm{nd}\;i}=1-\alpha -\Phi^{\prime }\left(\Phi^{-1}\left(1-\alpha \right)\right)\cdot \left(\Phi^{-1}\left(1-\alpha \right)+\frac{\sqrt{N_{F}N_{H}}}{\sqrt{2}}\right)\frac{\sigma_{s}^{2}\left(i\right)}{\sigma_{\xi }^{2}\left(i\right)}. \end{array}$$


**Proof.** Choose the non-detection probability of the form Equation (Equation 10) and consider it at a small signal/noise ratio
(12)$$\begin{array}{c} P_{\mathrm{nd}}=\Phi \left(\frac{\sigma_{\xi }^{2}\Phi^{-1}\left(1-\alpha \right)}{\sigma_{s}^{2}+\sigma_{\xi }^{2}}-\frac{\sigma_{s}^{2}\sqrt{N_{F}N_{H}}}{\left(\sigma_{s}^{2}+\sigma_{\xi }^{2}\right)\sqrt{2}}\right)=\Phi \left(\Phi^{-1}\left(1-\alpha \right)-\left(\Phi^{-1}\left(1-\alpha \right)+\frac{\sqrt{N_{F}N_{H}}}{\sqrt{2}}\right)\frac{\sigma_{s}^{2}}{\sigma_{\xi }^{2}}+o\left(\frac{\sigma_{s}^{2}}{\sigma_{\xi }^{2}}\right)\right). \end{array}$$Taylor series expansion with respect to a small signal/noise parameter leads Equation (Equation 12) to the form
$$\begin{array}{c} P_{\mathrm{nd}}=\Phi \left(\Phi^{-1}\left(1-\alpha \right)\right)-\Phi^{\prime }\left(\Phi^{-1}\left(1-\alpha \right)\right)\cdot \left(\Phi^{-1}\left(1-\alpha \right)+\frac{\sqrt{N_{F}N_{H}}}{\sqrt{2}}\right)\frac{\sigma_{s}^{2}}{\sigma_{\xi }^{2}}+o\left(\frac{\sigma_{s}^{2}}{\sigma_{\xi }^{2}}\right), \end{array}$$
which is simplified to an expression $P_{\mathrm{nd}}=1-\alpha -\Phi^{\prime }\left(\Phi^{-1}\left(1-\alpha \right)\right)\cdot \left(\Phi^{-1}\left(1-\alpha \right)+\frac{\sqrt{N_{F}N_{H}}}{\sqrt{2}}\right)\frac{\sigma_{s}^{2}}{\sigma_{\xi }^{2}}+o\left(\frac{\sigma_{s}^{2}}{\sigma_{\xi }^{2}}\right).$ □

The total probability of UUV detection formally corresponds to the probability of occurrence of a detection event at least once in a series of *N* observations and for independent events (observations) is calculated by the formula $P_{d}=1-\prod_{i=1}^{N}P_{\mathrm{nd}\;i},$ or, if you introduce the concept of “risk”,
(13)$$R=\sum_{i=1}^{N}R_{i},$$
where $R_{i}=-\ln P_{\mathrm{nd}\;i}$ is the risk for the *i*-th observation segment. Given Equation (Equation 12), for enough small α Formula (Equation 13) takes the form
$$\begin{array}{c} R=\alpha N+\Phi^{\prime }\left(\Phi^{-1}\left(1-\alpha \right)\right)\cdot \left(\Phi^{-1}\left(1-\alpha \right)+\frac{\sqrt{N_{F}N_{H}}}{\sqrt{2}}\right)\cdot \sum_{i=1}^{N}{\frac{\sigma_{s}^{2}\left(i\right)}{\sigma_{\xi }^{2}\left(i\right)}}_{}. \end{array}$$

Multiplying and dividing the last expression by ΔT (tact duration) and assuming that $T_{0}=N\Delta T$ (UUV time on the route), we obtain
(14)$$\begin{array}{c} R=\frac{\alpha T_{0}}{\Delta T}+\Phi^{\prime }\left(\Phi^{-1}\left(1-\alpha \right)\right)\cdot \left(\Phi^{-1}\left(1-\alpha \right)+\frac{\sqrt{N_{F}N_{H}}}{\sqrt{2}}\right)\cdot \frac{1}{\Delta T}\sum_{i=1}^{N}\frac{\sigma_{s}^{2}\left(i\right)}{\sigma_{\xi }^{2}\left(i\right)}\Delta T. \end{array}$$

The last sum is represented as an integral, which we call the threat functional [4,5] in the problem of UUV route planning for the case of passive sonar
(15)$$R=\int_{0}^{T_{0}}\frac{\sigma_{s}^{2}\left(t\right)}{\sigma_{\xi }^{2}\left(t\right)}dt.$$

This conclusion coincides with the result given in Reference [3] in the case of one sonar for
(16)σs2=σ02υjυjυ0υ0μr02r2γandσξ2(t)=const.
with the non-detection probability in the form
(17)Pnd=∏jFnχ1−α,ξ21+σ02υjυjυ0υ0μr02σξ2rj2γ,
where $F_{n}\;\;\mathrm{is}\;\;\chi^{2}$ – probability distribution with *n* degrees of freedom, α is a false alarm probability, γ– attenuation coefficient, $r_{j}$ is the distance from UUV to sonar at *j* tact, starting from the moment of appearance of UUV on the trajectory, and $\upsilon_{j}$ is a constant speed of UUV at this tact, $\sigma_{0}$, $\sigma_{\xi }$, μ, $r_{0}$, $\upsilon_{0}$ are some model parameters. The number of factors in the product is equal to the number *N* of tacts when moving UUV along the trajectory.

To use the Formula (Equation 14) for calculations, it is necessary to ensure the smallness of the first term and to estimate the value of the multiplier standing before the sign of the sum in the second term. The dependence $G\left(\alpha \right)=\Phi^{\prime }\left(\Phi^{-1}\left(1-\alpha \right)\right)\cdot \left(\Phi^{-1}\left(1-\alpha \right)+\frac{1}{\sqrt{2}}\right)$ of this multiplier on the probability of a false alarm is shown in Figure 1. Moreover, for the path planning task, it is not required to know the exact values of the information processing parameters included in expression Equation (Equation 14) because they are not included in Equation (Equation 15).

### 2.4. Non-Detection Probability of UUV under Active Surveillance

Consider the case of an active location, when the observation model has the form
$$x\left(f\right)=\delta \cdot s\left(f\right)+\xi \left(f\right),\;\;f=1,\ldots ,N_{F},$$
$$D\left[Q|H_{0}\right]=D\left[Q|H_{1}\right]=4s^{*}\left(f\right)K_{\xi }^{-1}\left(f\right)s\left(f\right),$$
where x(f), s(f), ξ(f) are Gaussian complex input signal, useful signal and noise, respectively, after FFT conversion. The meaning of δ is the same. In this case, the statistics for comparing with the threshold is as follows
(18)$$Q=\sum_{f=1}^{N_{F}}\;\left[s^{*}\left(f\right)K_{\xi }^{-1}\left(f\right)x\left(f\right)+x_{}^{*}\left(f\right)K_{\xi }^{-1}\left(f\right)s\left(f\right)\right],$$
where $K_{\xi }$ is the covariance matrix of noise vector. Statistics Equation (Equation 18) is linear in observations, $P(T\leq h|H_{1})$ has a normal distribution and the non-detection probability on the observation cycle after sending the probing signal is written in the form of Equation (Equation 8), but here
$$E\left[Q|H_{0}\right]=0,\;\;E\left[Q|H_{1}\right]=2s^{*}\left(f\right)K_{\xi }^{-1}\left(f\right)s\left(f\right),$$
and hence
$$P\left(Q\leq h|H_{1}\right)=\Phi \left(\Phi^{-1}\left(1-\alpha \right)-\frac{1}{\sqrt{s^{*}\left(f\right)K_{\xi }^{-1}\left(f\right)s\left(f\right)}}\right).$$

In the absence of information about covariances between hydrophones the probability of non-detection at one sending of the probing signal is equal to
(19)P(Q≤h|H1)=ΦΦ−1(1−α)−1s*(f)s(f)s*(f)s(f)σn2σξ2.

The number of factors in the product is equal to the number of cycles during the passage of the entire trajectory of UV. As in Equation (Equation 11), the non-detection probability on the trajectory in active mode is the product of all observation cycles on the trajectory
(20)Pnd=∏jP(Qj≤h|H1)=∏jΦΦ−1(1−α)−1s*(f)s(f)s*(f)s(f)σn2σξ2.

It should be noted that in Formulas (Equation 11) and (Equation 20), the variance of the useful signal is a function not only of the distance *r* to the object, but also of the attenuation coefficient γ when the acoustic wave propagates in a stratified medium.

Choose the non-detection probability of the form Equation (Equation 20) and consider it at a small signal/noise ratio with the aid of Taylor series. Then next formula is valid. Like Formula (Equation 15), summing up all the cycles of observation, we obtain the treat functional part of active mode equal to
(21)$$R=\int_{0}^{T}\frac{\sigma_{\xi }^{}\left(t\right)}{\sqrt{s_{f}^{*}\left(t\right)s_{f}\left(t\right)}}dt.$$

The treat functional Equation (Equation 15) or (Equation 21) on a segment of the trajectory has as input parameters the data of sonar systems, including their location (coordinates and parameters of detection active or passive mode, orientation of antenna, etc.) and calculated for these values of the ratio signal/noise reference source (its speed and the direction of sonar), the values of latitude and longitude start and end points of the segment, the speed of the UUV.

## 3. Path Planing Problem

### 3.1. About Risk Functional

The path planning problem for object moving in a threat environment will be determined as the problem of calculus of variations of the risk functional Equation (Equation 15). Due to Equation (Equation 16), after omitting the constants the expression under integral depends on instantaneous level of signal S(t), received by the sonar [1]
$$\frac{\sigma_{s}^{2}\left(t\right)}{\sigma_{\xi }^{2}\left(t\right)}∼S\left(t\right).$$

This signal itself depends on the receiving diagram of the sensor and the radiation pattern of the object as well as properties of the marine environment
(22)$$S=\frac{v^{\mu }}{r^{k}}G\left(\varphi \right)\gamma g\left(\beta \right),$$
where multiplier G(φ) is responsible for diagram of sensor antenna, g(β) is a radiation pattern of the moving object, *r* is the current distance between sensor and evading object, *v* – absolute instant velocity of the object, γ – signal saturation factor in water. The geometric meaning of angles ψ and φ is as follows. Here ψ is the angle of rotation of object’s velocity vector and φ is the angle of rotation of radius vector as shown in Figure 2. Angle ψ−φ=β is the angle in triangle built from radial $v_{r}$ and transversal $v_{\varphi }$ velocities of the object, for example, the angle between velocity of the object and its projection on radius vector as shown in Figure 3.The exponents *k* and μ characterize the physical field used for detection. Depending on values of *k* and μ this can be magnetic, thermal, acoustic or electromagnetic fields.

Moreover, the risk *R* is the integral value of this signal, so the criterion of optimization Equation (Equation 15) is a function of phase coordinates of the object and qualities of the sonar and the object itself:(23)$$R=\int_{0}^{T_{0}}\left(\frac{v^{\mu }}{r^{k}}G\left(\varphi \right)\gamma g\left(\beta \right)\right)dt.$$

We consider sonar antenna diagram to be homogeneous, so G(φ)≡1, and there is no additional signal attenuation or gain in the environment so γ=1.

We study the case of k=μ, namely μ=2 for acoustic field in underwater environment. Non-uniform radiation pattern of the object can be described as
$$g\left(\beta \right)=F\left(\frac{\dot{\varphi }r}{\dot{r}}\right).$$

The example of radiation pattern is presented in Figure 4.

### 3.2. Mathematical Statement of the Problem

Therefore we can formulate the optimal path planning problem.

**Problem** **1.**
*It is required to find such trajectory $\left(r^{*}\left(t\right),\varphi^{*}\left(t\right)\right)$, which minimizes the functional*
(24)$$R=\int_{0}^{T_{0}}{\left(\frac{v}{r}\right)}^{\mu }F\left(\frac{\dot{r\varphi }}{\dot{r}}\right)dt,$$
*where v is a velocity of the object, r is the distance between the sensor and the object. Boundary conditions are*
$$r\left(0\right)=x_{A},\;r\left(T_{0}\right)=r_{B},\;\varphi \left(0\right)=\varphi_{A},\;\varphi \left(T_{0}\right)=\varphi_{B}.$$

*Time $T_{0}$ of moving on route from point A to B is fixed. The detection system consists of one static sonar placed in the origin of Cartesian coordinate system with X axis passing through point A. The polar coordinate system is used for solution simplicity.*


### 3.3. Solution of the Path Planing Problem

**Lemma** **2.**
*Substitution of variable ρ=lnr brings functional Equation (Equation 24) to the form*
(25)$$R\left(\rho ,\dot{\rho },\varphi ,\dot{\varphi },t\right)=\int_{0}^{T_{0}}{\left({\dot{\rho }}^{2}+{\dot{\varphi }}^{2}\right)}^{\mu /2}F\left(\frac{\dot{\varphi }}{\dot{\rho }}\right)dt.$$


Proof of lemma 2 is given in the Appendix A.

Instead of solving the original Problem 1 according to Lemma 2 it is necessary to solve a two point boundary value variational problem on the minimum of the functional Equation (Equation 25).

**Problem** **2.**
*It is required to find the trajectory $\left(\rho^{*}\left(t\right),\varphi^{*}\left(t\right)\right)$, which minimizes the functional*
(26)$$R\left(\rho ,\dot{\rho },\varphi ,\dot{\varphi },t\right)=\int_{0}^{T_{0}}S\left(\rho ,\dot{\rho },\varphi ,\dot{\varphi },t\right)dt=\int_{0}^{T_{0}}{\left({\dot{\rho }}^{2}+{\dot{\varphi }}^{2}\right)}^{\mu /2}F\left(\frac{\dot{\varphi }}{\dot{\rho }}\right)dt⟶\min_{\rho (\cdot ),\varphi (\cdot )}.$$

*with boundary conditions*
$$\rho \left(0\right)=\rho_{A},\;\rho \left(T_{0}\right)=\rho_{B},\;\varphi \left(0\right)=\varphi_{A},\;\varphi \left(T_{0}\right)=\varphi_{B}.$$


The following lemma holds.

**Lemma** **3.**
*The lagrangian $S(\rho ,\dot{\rho },\varphi ,\dot{\varphi },t)$ has a constant value $S^{*}$ along the extremal trajectory $\left(\rho^{*}\left(t\right),\varphi^{*}\left(t\right)\right)$.*


**Proof.** Indeed, function $S\left(\rho ,\varphi ,\dot{\rho },\dot{\varphi },t\right)={\left({\dot{\rho }}^{2}+{\dot{\varphi }}^{2}\right)}^{\mu /2}F\left(\eta \right)$, where $\eta =\frac{\dot{\varphi }}{\dot{\rho }}$, does not explicitly depend on variable *t*. Hence the value of generalized energy $H=\dot{\rho }\frac{\partial S}{\partial \dot{\rho }}+\dot{\varphi }\frac{\partial S}{\partial \dot{\varphi }}-S$ is a constant along the extremal trajectory $\left(\rho^{*}\left(t\right),\varphi^{*}\left(t\right)\right)$. The calculation in the last expression gives a chain of equalities
$$\begin{array}{c} H=\dot{\rho }\dot{\rho }\mu {\left({\dot{\rho }}^{2}+{\dot{\varphi }}^{2}\right)}^{\mu /2-1}F\left(\eta \right)+\dot{\varphi }\dot{\varphi }\mu {\left({\dot{\rho }}^{2}+{\dot{\varphi }}^{2}\right)}^{\mu /2-1}F\left(\eta \right)+\\ +\dot{\rho }{\left({\dot{\rho }}^{2}+{\dot{\varphi }}^{2}\right)}^{\mu /2}F^{\prime }\left(\eta \right)\left(-\frac{\dot{\varphi }}{{\dot{\rho }}^{2}}\right)+\dot{\varphi }{\left({\dot{\rho }}^{2}+{\dot{\varphi }}^{2}\right)}^{\mu /2}F^{\prime }\left(\eta \right)\left(\frac{1}{\dot{\rho }}\right)-\\ -{\left({\dot{\rho }}^{2}+{\dot{\varphi }}^{2}\right)}^{\mu /2}F\left(\eta \right)=\left(\mu -1\right)S=\left(\mu -1\right)S^{*}. \end{array}$$In the case of μ=1 the problem degenerates and requires additional exploration. □

**Theorem** **1.**
*Suppose that $0<F_{1}<F\left(\eta \right)<F_{2}$ for all $\eta \in R^{1}$ is a twice continuously differentiated function of η, where $F_{1}$, $F_{2}$ are some constant values, and $\ddot{\rho }\left(t\right)$, $\ddot{\varphi }\left(t\right)$ exist and are continuous functions of t. Then the extremal trajectory satisfies the following system of equations*
(27)$$\left\{\begin{array}{c} \dot{\rho }=const,\\ \dot{\varphi }=const. \end{array}\right.$$


**Proof.** The Euler-Lagrange equations for the variational problem Equation (Equation 26) are
(28)$$\left\{\begin{array}{c} -\frac{d}{dt}\frac{\partial S}{\partial \dot{\rho }}\left(\rho ,\varphi ,\dot{\rho },\dot{\varphi },t\right)+\frac{\partial S}{\partial \rho }\left(\rho ,\varphi ,\dot{\rho },\dot{\varphi },t\right)=0,\\ -\frac{d}{dt}\frac{\partial S}{\partial \dot{\varphi }}\left(\rho ,\varphi ,\dot{\rho },\dot{\varphi },t\right)+\frac{\partial S}{\partial \varphi }\left(\rho ,\varphi ,\dot{\rho },\dot{\varphi },t\right)=0, \end{array}\right.$$
where $S\left(\rho ,\varphi ,\dot{\rho },\dot{\varphi },t\right)={\left({\dot{\rho }}^{2}+{\dot{\varphi }}^{2}\right)}^{\mu /2}F\left(\frac{\dot{\varphi }}{\dot{\rho }}\right)$.After substituting the functional *S* into system Equation (Equation 28) and due to $\frac{\partial S}{\partial \varphi }=0$, $\frac{\partial S}{\partial \rho }=0$, we have
(29)$$\left\{\begin{array}{c} \mu \dot{\rho }{\left({\dot{\rho }}^{2}+{\dot{\varphi }}^{2}\right)}^{\mu /2-1}F\left(\eta \right)-{\left({\dot{\rho }}^{2}+{\dot{\varphi }}^{2}\right)}^{\mu /2}F^{\prime }\left(\eta \right)\frac{\dot{\varphi }}{{\dot{\rho }}^{2}}=C_{1},\\ \mu \dot{\varphi }{\left({\dot{\rho }}^{2}+{\dot{\varphi }}^{2}\right)}^{\mu /2-1}F\left(\eta \right)+{\left({\dot{\rho }}^{2}+{\dot{\varphi }}^{2}\right)}^{\mu /2}F^{\prime }\left(\eta \right)\frac{1}{\dot{\rho }}=C_{2}. \end{array}\right.$$Here $C_{1}$, $C_{2}$ are some constant values, which are first integrals of system Equation (Equation 28) as well as $S^{*}$. After multiplying first Equation (Equation 29) on $\dot{\rho }$, second – on $\dot{\varphi }$ and summing, so on after multiplying first Equation (Equation 29) on $-\dot{\rho }$, second – on $\dot{\varphi }$ and summing, we have system
(30)$$\left\{\begin{array}{c} C_{1}\dot{\rho }+C_{2}\dot{\varphi }=\mu S^{*},\\ C_{2}\dot{\rho }-C_{1}\dot{\varphi }=F^{\prime }\left(\eta \right)\left(\eta^{2}+1\right){\left({\dot{\rho }}^{2}+{\dot{\varphi }}^{2}\right)}^{\mu /2}. \end{array}\right.$$The implementation the condition of theorem $F>F_{1}>0$ and substitution the $S^{*}$ into second Equation (Equation 30) give
(31)$$\left\{\begin{array}{c} C_{1}\dot{\rho }+C_{2}\dot{\varphi }=\mu S^{*},\\ C_{2}\dot{\rho }-C_{1}\dot{\varphi }={\left(\ln F(\eta )\right)}^{\prime }\left(\eta^{2}+1\right)S^{*}. \end{array}\right.$$Dividing the second equation from Equation (Equation 31) on the first and substituting $\eta =\frac{\dot{\varphi }}{\dot{\rho }}$ the equation with respect to variable η one can obtain
(32)$${\left(\ln F(\eta )\right)}^{\prime }=\frac{C_{2}-C_{1}\eta }{C_{1}+C_{2}\eta }\frac{\mu }{\eta^{2}+1}.$$It means that left and right sides of Equation (Equation 32) equal to each other at some fixed points $\eta^{*}$. From the linearity on $C_{1}$, $C_{2}$ system Equation (Equation 31) follows its solvability at any right-hand side. Due to fixed value $\eta^{*}$ and left side of Equation (Equation 31) the statement of theorem follows. □

The general form of optimal trajectory can be obtained.

**Lemma** **4.**
*Let $S\left(\rho ,\dot{\rho },\varphi ,\dot{\varphi },t\right)=\left({\dot{\rho }}^{2}+{\dot{\varphi }}^{2}\right)g\left(\beta \left(\eta \right)\right)$, g(β) – thrice continuously differentiated function of β, then Hessian matrix H equals*
(33)$$H=\;\left(\begin{array}{cc} 2g\left(\beta \right)-2g^{\prime }\left(\beta \right)\frac{\dot{\varphi }\dot{\rho }}{{\dot{\rho }}^{2}+{\dot{\varphi }}^{2}}+g^{\prime \prime }\left(\beta \right)\frac{{\dot{\varphi }}^{2}}{{\dot{\rho }}^{2}+{\dot{\varphi }}^{2}} & g^{\prime }\left(\beta \right)\frac{{\dot{\rho }}^{2}-{\dot{\varphi }}^{2}}{{\dot{\rho }}^{2}+{\dot{\varphi }}^{2}}-g^{\prime \prime }\left(\beta \right)\frac{\dot{\varphi }\dot{\rho }}{{\dot{\rho }}^{2}+{\dot{\varphi }}^{2}},\\ g^{\prime }\left(\beta \right)\frac{{\dot{\rho }}^{2}-{\dot{\varphi }}^{2}}{{\dot{\rho }}^{2}+{\dot{\varphi }}^{2}}-g^{\prime \prime }\left(\beta \right)\frac{\dot{\varphi }\dot{\rho }}{{\dot{\rho }}^{2}+{\dot{\varphi }}^{2}} & 2g\left(\beta \right)+2g^{\prime }\left(\beta \right)\frac{\dot{\varphi }\dot{\rho }}{{\dot{\rho }}^{2}+{\dot{\varphi }}^{2}}+g^{\prime \prime }\left(\beta \right)\frac{{\dot{\rho }}^{2}}{{\dot{\rho }}^{2}+{\dot{\varphi }}^{2}} \end{array}\right)$$

*with*
(34)$$\det H=4g^{2}\left(\beta \right)+2g\left(\beta \right)g^{\prime \prime }\left(\beta \right)-g^{\prime 2}\left(\beta \right).$$


**Proof.** After calculating the first partial derivatives of *S* on $\dot{\rho }$, $\dot{\varphi }$, using equality $\beta =\arctan\left(\frac{\dot{\varphi }}{\dot{\rho }}\right)$, one can get
(35)$$\begin{array}{c} S_{\dot{\rho }}=2\dot{\rho }g\left(\beta \right)-\dot{\varphi }g^{\prime }\left(\beta \right),\\ S_{\dot{\varphi }}=2\dot{\varphi }g\left(\beta \right)+\dot{\rho }g^{\prime }\left(\beta \right). \end{array}$$Then we can calculate the second partial derivatives of *S* on $\dot{\rho }$, $\dot{\varphi }$ and obtain Equation (Equation 33). Computing the determinant of the Hessian matrix gives Equation (Equation 34). □

**Theorem** **2.**
*Let us suppose that conditions of theorem 1, lemmas 3 and 4 is fulfilled, and det H>0 for all values β. The optimal trajectory satisfying Equation (Equation 27) gives the strong minimum to the risk functional Equation (Equation 25).*


**Proof.** First of all, it is required to check the Legandre conditions. It means that Hessian matrix is positively defined. Indeed second order minor coincides with detH which is strictly positive. Among first order minors $H_{11}$ and $H_{22}$ at list one of them is positive because both values are fixed on the optimal trajectory and their sum equals
$$H_{11}+H_{22}=4g\left(\beta \right)+g^{\prime \prime }\left(\beta \right)=\frac{4g^{2}\left(\beta \right)+\det H+g^{\prime 2}\left(\beta \right)}{2g(\beta )}>0.$$Hence both values $H_{11}$ and $H_{22}$ are strictly positive too.The Jacobin conditions have the form
(36)$$\begin{array}{c} \frac{d}{dt}\left(H_{11}\dot{\rho }+H_{12}\dot{\varphi }\right)=0,\\ \frac{d}{dt}\left(H_{21}\dot{\rho }+H_{22}\dot{\varphi }\right)=0, \end{array}$$
because $S_{\rho \dot{\rho }}=0$, $S_{\varphi \dot{\rho }}=0$, $S_{\rho \dot{\varphi }}=0$, $S_{\varphi \dot{\varphi }}=0$. The Equation (Equation 36) take the form
$$\begin{array}{c} \frac{d}{dt}S_{\dot{\rho }}=0,\\ \frac{d}{dt}S_{\dot{\varphi }}=0, \end{array}$$
which coincides with explicit form of Equation (Equation 28) and leads to Equation (Equation 29). It means that the sufficient conditions of strong minimum of risk functional Equation (Equation 25) is fulfilled [23]. □

**Corollary** **1.**
*According to the theorems 1 and 2 the optimal trajectory has the form of logarithmic spiral.*
(37)$$\left\{\begin{array}{c} r\left(t\right)=r_{A}\exp\left(\frac{t}{T_{0}}\ln\frac{r_{B}}{r_{A}}\right),\\ \varphi \left(t\right)=\varphi_{A}+\frac{\varphi_{B}-\varphi_{A}}{T_{0}}t. \end{array}\right.$$

*The equation of optimal trajectory of the UUV on the plane is*
(38)$$r\left(\varphi \right)=r_{A}\exp\left(\frac{\varphi -\varphi_{A}}{\varphi_{B}-\varphi_{A}}\ln\frac{r_{B}}{r_{A}}\right).$$


**Lemma** **5.**
*The functional value Equation (Equation 25) on the optimal trajectory Equation (Equation 38) equals*
(39)$$R^{*}=\frac{1}{T_{0}}\left({\left(\rho_{B}-\rho_{A}\right)}^{2}+{\left(\varphi_{B}-\varphi_{A}\right)}^{2}\right)g\left(\beta^{*}\right),$$
*where $\beta^{*}=\arctan\frac{\varphi_{B}-\varphi_{A}}{\rho_{B}-\rho_{A}}$, and depends on boundary conditions only.*


The proof of lemma is given in Appendix A.

The theorems above guarantee that optimal trajectory of the object is logarithmic spiral. However, it is unclear, whether this trajectory must consist of one segment or several and have points of switching along the route. To answer this question let us compare the values of the risk functional in two cases. In the first, object moves from *A* straight to *B*, and in second - from *A* to *C* and then to *B*. In first case risk $R_{AB}^{*}$, according to Lemma 5, equals
(40)$$R_{AB}^{*}=\frac{{\left(\rho_{B}-\rho_{A}\right)}^{2}+{\left(\varphi_{B}-\varphi_{A}\right)}^{2}}{T_{0}}g\left(\arctan\left(\frac{\varphi_{B}-\varphi_{A}}{\rho_{B}-\rho_{A}}\right)\right).$$

The risk on the trajectory consisting of two segments is given in the next Lemma.

**Lemma** **6.**
*Let g(β)=g(−β)=g(π−β) for all 0≤β≤2π. The minimal value of risk on the two-segment trajectory consequently passing through the points A, C, B during time interval $T_{0}$ equals*

*a) in the case of $0\leq \beta_{m}<\beta^{*}$*
(41)$$R_{AC}^{*}+R_{CB}^{*}=\frac{{\left(\rho_{B}-\rho_{A}\right)}^{2}}{T_{0}\cos^{2}\beta_{m}}g\left(\beta_{m}\right),$$

*b) in the case of $\beta^{*}<\beta_{m}\leq \pi /2$*
(42)$$R_{AC}^{*}+R_{CB}^{*}=\frac{{\left(\varphi_{B}-\varphi_{A}\right)}^{2}}{T_{0}\sin^{2}\beta_{m}}g\left(\beta_{m}\right),$$

*c) in the case of $\beta_{m}=\beta^{*}$*
(43)$$R_{AC}^{*}+R_{CB}^{*}=R_{AB}^{*}=\frac{{\left(\rho_{B}-\rho_{A}\right)}^{2}}{T_{0}\cos^{2}\beta^{*}}g\left(\beta^{*}\right)=\frac{{\left(\varphi_{B}-\varphi_{A}\right)}^{2}}{T_{0}\sin^{2}\beta^{*}}g\left(\beta^{*}\right),$$

*where $\beta_{m}=\arctan\frac{\varphi_{C}-\varphi_{A}}{\rho_{C}-\rho_{A}}=-\arctan\frac{\varphi_{B}-\varphi_{C}}{\rho_{B}-\rho_{C}}$ and*
$$\begin{array}{c} \rho_{C}=\frac{\rho_{A}+\rho_{B}+\left(\varphi_{B}-\varphi_{A}\right)\cot\beta_{m}}{2},\\ \varphi_{C}=\frac{\varphi_{A}+\varphi_{B}+\left(\rho_{B}-\rho_{A}\right)\tan\beta_{m}}{2}. \end{array}$$


The proof of Lemma 6 is given in Appendix A.

Due to the fact that real acoustic radiation patterns of UUV have more complex shapes, Lemma 6 provides a simple way to check the type of optimal trajectory. Indeed, if in one of the cases of Lemma 6, $R_{AC}^{*}+R_{CB}^{*}<R_{AB}^{*}$, then the logarithmic spiral directly connecting the points *A*, *B* is not optimal and two-segment trajectory is preferable in a sense of risk functional value.

## 4. Discussion of the Results

In previous articles the similar problem without radiation pattern was considered. The risk functional has a form
$$R=\int_{0}^{T_{0}}\frac{v^{\mu }}{r^{2}}dt\to \min_{\left(v(\cdot ),r(\cdot )\right)}.$$

In Reference [1], it was proven that optimal trajectory and velocity in the case of μ=2 for underwater environment, considered in the current article, are the same as Equation (Equation 38). Generally speaking, this is a non-trivial fact. However, the optimal risk value in Reference [1] equals
$$R=\frac{1}{T_{0}}\left({\left(\rho_{B}-\rho_{A}\right)}^{2}+{\left(\varphi_{B}-\varphi_{A}\right)}^{2}\right).$$

Comparing this formula with Equation (Equation 39), one can see that taking into account heterogeneous radiation pattern of the moving object reduces risk value, as g(β), being normalized for comparison by the meaning max(g(β)), is less than 1. Then the risk value Equation (Equation 39) is less than the one in Reference [1].

Matlab procedure has been developed to simulate and validate found analytical solution. Figure 5 and Figure 6 show results of this simulation. In each figure left picture shows the chosen radiation pattern and right picture shows extremal trajectories, obtained using this pattern: path $AC_{1}B$ represents case a) of the Lemma 6, $AC_{2}B$ – case b) and AB – case c) respectively. As a result numerical and analytical solutions have been found to coincide with each other that supports Lemma 6.

The values of risk functional and time intervals have been calculated. In the first example for case a) $T_{1}=8.8566$, $R_{AC_{1}}^{*}=0.078529$, $T_{2}=11.1434$, $R_{C_{1}B}^{*}=0.062401$, $R_{AC_{1}B}^{*}=0.13909$, for case b) $T_{1}=11.1434$, $R_{AC_{2}}^{*}=0.078529$, $T_{2}=8.8566$, $R_{C_{2}B}^{*}=0.067223$, $R_{AC_{2}B}^{*}=0.13909$, for case c) $R_{AB}^{*}=0.14739$. One can see that $R_{AC_{1}B}^{*}=R_{AC_{2}B}^{*}$. Also it is clear that $R_{AC_{1}}^{*}=R_{C_{2}B}^{*}$ and $R_{C_{1}B}^{*}=R_{AC_{2}}^{*}$ due to geometry of placement points $C_{1}$, $C_{2}$ on plane, which are defined by appropriate choice of radiation pattern.

This is also fair for the second example, where for case a): $T_{1}=6.7286$, $R_{AC_{1}}=0.079755$, $T_{2}=13.2714$, $R_{C_{1}B}^{*}=0.062401$, $R_{AC_{1}B}^{*}=0.17911$, for case b): $T_{1}=13.2714$, $R_{AC_{2}}^{*}=0.10141$, $T_{2}=6.7286$, $R_{C_{2}B}^{*}=0.079755$, $R_{AC_{2}B}^{*}=0.17911$, for case c): $R_{AB}^{*}=0.19675$.

In both examples, $R_{AB}^{*}>R_{AC_{1}B}^{*}$, that is, two-segment route has a less risk of detection than the one of the simple straight route. Thus this solution is different from results of Reference [1] and presents an advantage in the sense of risk functional minimization. Whereas in Reference [1], due to velocity optimization risk value on the optimal path was reduced by 40% in average, now due to radiation pattern risk has been reduced by factor $\frac{g(\beta^{*})}{\max(g(\beta ))}$ up to value $\frac{\min(g(\beta ))}{\max(g(\beta ))}$. In both examples the maximum primary acoustic radiation of UUV while object moves along the trajectory AB falls in the direction of SSS or, in other words, $g\left(\beta^{*}\right)=\max\left(g\left(\beta \right)\right)$. Therefore, under other boundary conditions, the risk functional reduction will be significantly greater. This presents an advantage of new solution in the sense of risk functional minimization. Moreover, in both examples $R_{AB}^{*}>R_{AC_{1}B}^{*}$, that is, two-segment route has a less risk of detection than the one of the simple straight route (risk reduction is by 6% and 8% respectively), that is the result described analytically in Lemma 6.

Now let us suppose that non-uniform radiation pattern of the object is described as
$$g\left(\beta \right)=K_{1}+K_{2}\cos^{2}\beta .$$

Coefficients $K_{1}$ and $K_{2}$ define the character of radiation and $K_{1}+K_{2}=1$. The example of this radiation pattern is presented in Figure 4 for $K_{1}=0.25,\;K_{2}=0.75$. The functional value Equation (Equation 25) on the optimal trajectory equals
$$R^{*}=\frac{1}{T_{0}}\left(\ln^{2}\frac{r_{B}}{r_{A}}+K_{1}^{2}{\left(\varphi_{B}-\varphi_{A}\right)}^{2}\right)$$
and depends on boundary conditions only. In this case according to Lemma 4 $\det H=4K_{1}\left(K_{1}+K_{2}\right)=4K_{1}>0$ and Theorem 2 can be reformulated as follows.

**Theorem** **3.**
*The optimal trajectory in the form Equation (Equation 37) gives the global minimum to the risk functional Equation (Equation 25).*


**Proof.** Indeed, the variation of functional Equation (Equation 25) has a form
$$\begin{array}{c} R\left(\rho +\delta \rho ,\dot{\rho }+\dot{\delta \rho },\varphi +\delta \varphi ,\dot{\varphi }+\dot{\delta \varphi },t\right)-R\left(\rho ,\dot{\rho },\varphi ,\dot{\varphi },t\right)=\\ =2\dot{\rho }\delta \rho \left|{}_{0}^{T_{0}}+\right.\dot{\varphi }\delta \varphi \left|{}_{0}^{T_{0}}\right.-2\int_{0}^{T_{0}}\left(\ddot{\rho }\delta \rho +\ddot{\varphi }\delta \varphi \right)dt+\\ +\int_{0}^{T_{0}}\left(\delta \rho^{2}+K_{1}\delta \varphi^{2}\right)dt=\int_{0}^{T_{0}}\left(\delta \rho^{2}+K_{1}\delta \varphi^{2}\right)dt>0, \end{array}$$
where δρ, δφ are variations of respective coordinates and smooth enough functions of time. The first and second summands are zeros due to fixed boundary conditions for variations, the third summand equals zero due to fulfilled Euler-Lagrange equations. The rest is positive as an integral of a weighted sum of squares of functions. Hence any variation of optimal trajectory enlarges the value of risk functional. □

## 5. Conclusions

Above, for simplicity, it was assumed that in the detection process of a UUV in passive mode using a linear antenna, bearing θ on the UUV is known. In fact, the bearing search for the target is carried out by additional optimization of statistics from observations, each of which depends on its direction θ to the target in the range (−π/2<θ<π/2). This means that the entire range of angles is divided into a finite number of angles (directions) $\theta_{m},m=1,\ldots M$. For each one it is necessary to solve the detection problem, that is, to build observation statistics depending on the bearing $\theta_{m}$ and then calculate and compare the corresponding detection probabilities. This procedure significantly increases the number of calculations. Therefore, in the future, an optimization method in the statistics space will be proposed, after which the probability of detection will be calculated only once.

When overcoming a particular network-centric system, the risk functional may consist of a weighted sum of the functions related to different search means. The UUV route planning problem in this case is reduced for example to a two-point variational problem or an optimal control problem in which the control variables enter into the permanent dispersion of the signal.

## Figures and Tables

**Figure 1 sensors-20-02076-f001:**
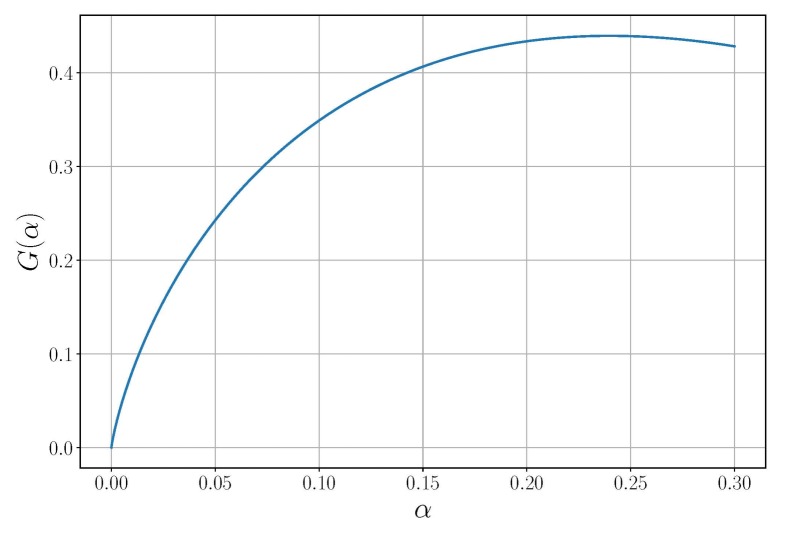
Dependence G(α).

**Figure 2 sensors-20-02076-f002:**
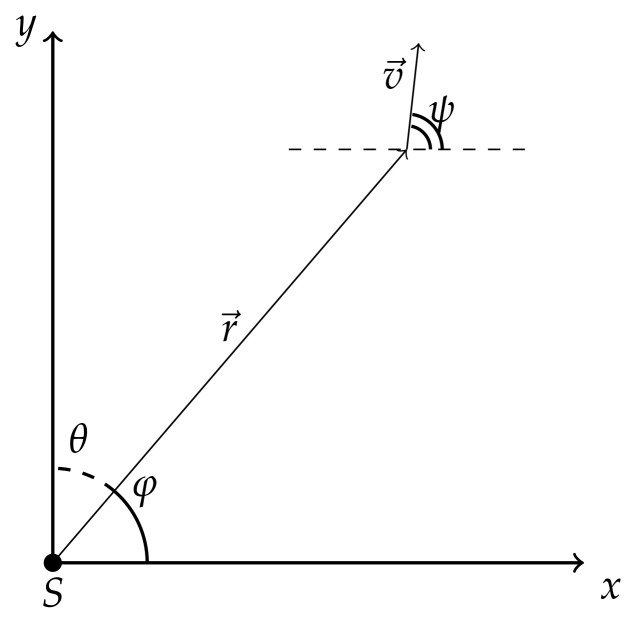
The moving object in the Cartesian coordinate system with sensor *S*.

**Figure 3 sensors-20-02076-f003:**
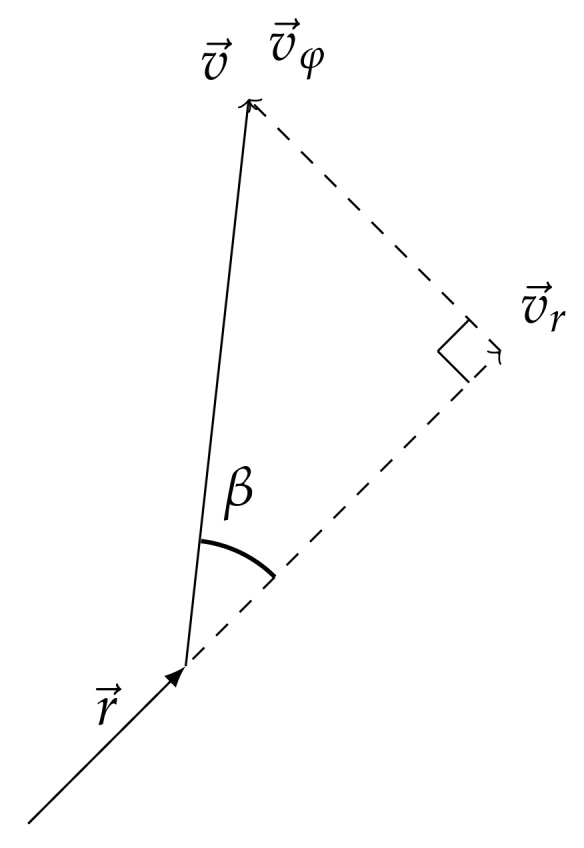
Rectangular triangle of velocity vectors $\vec{v},{\vec{v}}_{r}$ and ${\vec{v}}_{\varphi }$.

**Figure 4 sensors-20-02076-f004:**
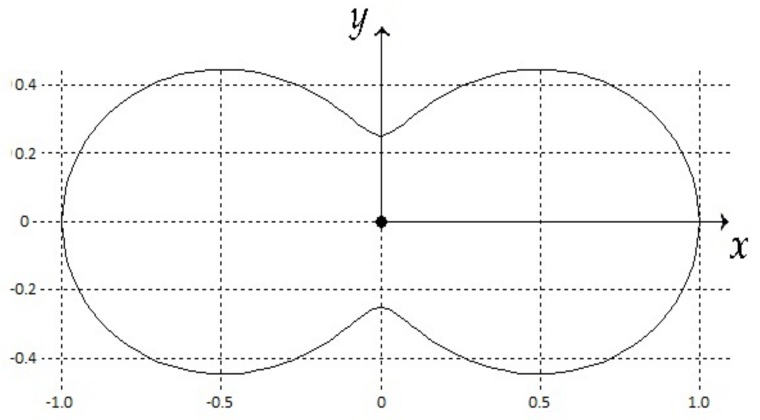
Radiation pattern of the object.

**Figure 5 sensors-20-02076-f005:**
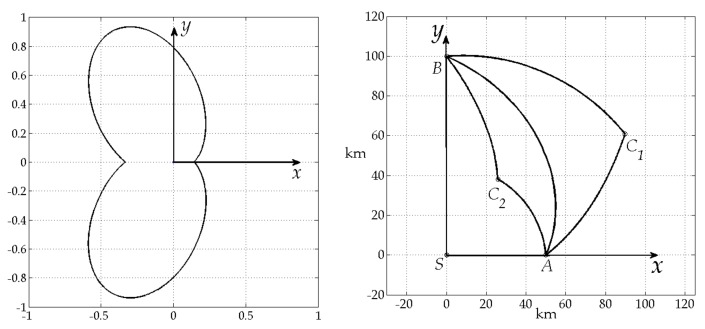
Radiation pattern **(left**) and extremal paths (**right**).

**Figure 6 sensors-20-02076-f006:**
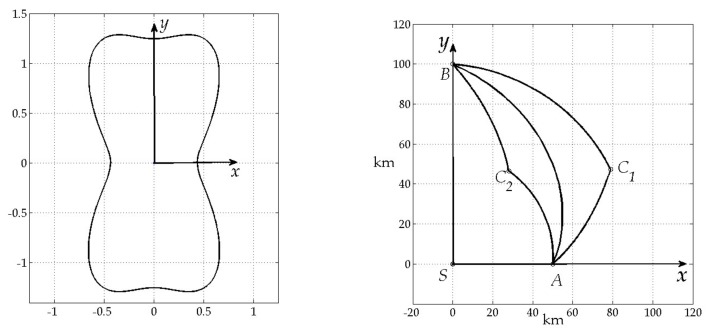
Radiation pattern (**left**) and extremal paths (**right**).

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
