# Peer review of "Path Planning in Threat Environment for UUV with Non-Uniform Radiation Pattern"

_sensors, 2020, doi:10.3390/s20072076_

Round 1

Reviewer 1 Report

Your paper explains originality solution for path planning problem of UAV with radiation pattern. However, I don't understand advantage of your method and results. In chapter 4 "Discussion of the results", you should explain simulation environment and show more simulation results for that redars understand merit of your method.

Author Response

Thank you very much for your revision.

Remark. However, I don't understand advantage of your method and results. In chapter 4 "Discussion of the results", you should explain simulation environment and show more simulation results for that redars understand merit of your method.

Answer. The advantage of our solution and results is given in paragraph in
section 4 "Discussion of the results", Page 16, starting from line
256. Another example has been added for illustrating Lemma 6 and risk
functional values have been given for comparison of different
solutions. Pattern figures have been added.

Moderate English changes have been made.

Reviewer 2 Report

This paper investigates the problem of optimal trajectory planning of the unmanned underwater vehicle (UUV). This problem is considered and analytically solved. The optimal trajectory and velocity law of the moving object are found, as well as the criterion value.

Generally speaking, the paper is pretty good. The technical contents are well written and well organized. This reviewer believes the paper is of merit and would like to recommend it for a possible acceptance, subjected to some minor revisions.

1 The references do not seem perfect. The authors are recommended to refer some latest papers.

2 The figure files are not good, i.e. Figures 3 and 5. Please make them clearer.

Author Response

Thank you very much for your review.

Remark 1. The references do not seem perfect. The authors are recommended to refer some latest papers.

Answer 1. Six new references for the period 2016-2020 have been added concerning the UUV path planning problems.

Remark 2. The figure files are not good, i.e. Figures 3 and 5. Please make them clearer.

Answer 2. Figures 3 and 5 have been redrawn without the gray background and the fonts have been changed.

Moderate English changes have been made.

Reviewer 3 Report

In this article, the authors have presented an analytical model to obtain the optimal trajectory of a UUV as well as the criterion value. However, it has some flaws need to be addressed.

  • Sometimes the article seems confusing when it talks about the operating environment. For instance, many times it refers to the distance between points in the space; however, the model is proposed for an underwater environment. 

  • The related work has not been investigated in this paper. The only related article is the reference (2) which has been proposed for UAV (Unmanned aerial vehicle) and not for UUV (Unmanned underwater vehicle). Authors need to add a related work section to explain the current works in the literature focusing on the underwater environment. 

  • The number of references is not enough and not covering the topic. Moreover, most of the references are old. 

  • Most of the claims in the article have not been referenced. For instance, in the introduction section, the first paragraph, the authors must provide appropriate references for each definition and application. In the fourth paragraph, there are many claims without even one reference. 

  • In lines 37-38, it has been mentioned that “Unlike previous works, the acoustic signal emitted by a moving object has a non-uniform pattern.” First, which previous works? Second, is there any reference for this claim in the literature?

  • In order to understand the application of the proposed model, the authors need to add an example with a figure to illustrate the model in the introduction section. 

  • At the end of the introduction section, the conclusion section (Section 5) should be referred to. 

  • Adding a flowchart would be helpful for the readers to understand the general idea of the article. 

  • All equations (taken from the literature) have not been cited and some of them have no numbering. 

  • In the result section, it is not clear how the physical layer of an underwater environment has been considered in the simulation. 

  • In line 229, it has been mentioned that “In previous articles, the similar problem without radiation pattern was considered.” In which previous work? 

  • There are also many grammatical and writing mistakes can be found throughout the article. Not using commas can cause confusion or even a misreading of the information.

  • In the abbreviations section, UUV is mistakenly considered as “Unmanned aerial vehicle” and UAV as “Unmanned underwater vehicle”.

Author Response

Thank you very much for your review.

Remark 1. Sometimes the article seems confusing when it talks about the operating environment. For instance, many times it refers to the distance between points in the space; however, the model is proposed for an underwater environment.

Answer 1. The coordinate system introduced in the article is connected with the antenna array of the sensor. In all formulas, the main distance r is measured from the origin to the center of a moving object.

Remark 2. The related work has not been investigated in this paper. The only related article is the reference (2) which has been proposed for UAV (Unmanned aerial vehicle) and not for UUV (Unmanned underwater vehicle). Authors need to add a related work section to explain the current works in the literature focusing on the underwater environment.

Remark 3. The number of references is not enough and not covering the topic. Moreover, most of the references are old.

Answer 2,3.  New six up-to-date references have been added:

Causa F., Fasano G., Grassi M.  Multi-UAV Path Planning for Autonomous Missions in Mixed GNSS Coverage Scenarios. {\em Sensors}  {\bf 2018}, 18, 4188.

Cited on page 1, line 33.

Guo S., Zhang X., Zheng Y., Du Y. An Autonomous Path Planning Model for Unmanned Ships Based on Deep Reinforcement Learning. {\em Sensors} {\bf 2020}, 20, 426.

Cited on page 2, line 35.

  1. Zhi-Wen, L. M. Kun and W. Li-jing Path Planning for UUV in Dynamic Environment. {\em 2016 9th International Symposium on Computational Intelligence and Design (ISCID)}, {\bf 2016}, 211--215.

Cited on page 2, line 35.

Cai W., Zhang M., Zheng Y.R. Task Assignment and Path Planning for Multiple Autonomous Underwater Vehicles Using 3D Dubins Curves. {\em Sensors} {\bf 2017}, 17, 1607.

Cited on page 2, line 36.

Panda, M., Das B., Subudhi B. et al. A Comprehensive Review of Path Planning Algorithms for Autonomous Underwater Vehicles. {\em Int. J. Autom. Comput.} {\bf 2020}.

Cited on page 2, line 38.

Ma S., Wang Y., Zou N., Liang G. A Broadband Beamformer Suitable for UUV to Detect the Tones Radiated from Marine Vessels. {\em Sensors} {\bf 2018}, 18, 2928.

Cited on page 2, line 44.

The distinctive features of these articles are discussed in the introduction.

Remark 4. Most of the claims in the article have not been referenced. For instance, in the introduction section, the first paragraph, the authors must provide appropriate references for each definition and application. In the fourth paragraph, there are many claims without even one reference.

Remark 5. In lines 37-38, it has been mentioned that “Unlike previous works, the acoustic signal emitted by a moving object has a non-uniform pattern.” First, which previous works? Second, is there any reference for this claim in the literature?

Answer 4,5. The additional citing references have been added with explanations.

Remark 6. In order to understand the application of the proposed model, the authors need to add an example with a figure to illustrate the model in the introduction section.

 Remark 8. Adding a flowchart would be helpful for the readers to understand the general idea of the article.

Answer 6,8. The last three paragraphs of the introduction discuss the structure of the work in detail. The main idea of which is to move from statistical hypothesis testing in the processing of hydroacoustic information to the problem of optimal path planning UUV. The authors think that according to the text of the article, figures appear in the right places and illustrate the essence of the work.

Remark 7. At the end of the introduction section, the conclusion section (Section 5) should be referred to.

Answer 7. The reference has been made.

Remark 9. All equations (taken from the literature) have not been cited and some of them have no numbering.

Answer 9. The main classical results are borrowed from the classical theory of hypothesis testing by Lehman and Monzingo, Miller, references to which are given in the list of references. Additional numbering has been made.

Remark 10. In the result section, it is not clear how the physical layer of an underwater environment has been considered in the simulation.

Remark 11. In line 229, it has been mentioned that “In previous articles, the similar problem without radiation pattern was considered.” In which previous work?

Answer 10,11. In this paper, the underwater environment was assumed to be uniform and stationary, since the main goal of the work is to account for the uneven radiation of UUV, i.e. to find out the influence of the uneven scattering indicatrix of an object on the criterion of its covertness. The advantage of our solution and results are given in section 4 "Discussion of the results", Page 15. Another example has been added for illustrating Lemma 6 and the risk functional values have been given for comparison of different solutions for UUV path planning problem.

Remark 12. There are also many grammatical and writing mistakes can be found throughout the article. Not using commas can cause confusion or even a misreading of the information.

Answer 12.  The authors have tried to correct this shortcoming throughout the text.

Remark 13. In the abbreviations section, UUV is mistakenly considered as “Unmanned aerial vehicle” and UAV as “Unmanned underwater vehicle”.

Answer 13.  Corrected.

Round 2

Reviewer 3 Report

Thanks to the authors who revised the manuscript. The paper can be accepted but it still needs another proofreading before publication.